# Surface Properties of 1DTiO_2_ Microrods Modified with Copper (Cu) and Nanocavities

**DOI:** 10.3390/nano11020324

**Published:** 2021-01-27

**Authors:** Snejana Bakardjieva, Filip Mamon, Zdenek Pinc, Radek Fajgar, Ivo Jakubec, Natalija Murafa, Eva Koci, Tatjana Brovdyova, Adriana Lancok, Stefan Michna, Rositsa Nikolova

**Affiliations:** 1Institute of Inorganic Chemistry of the Czech Academy of Sciences, 250 68 Rez, Czech Republic; mamon@iic.cas.cz (F.M.); jakubec@iic.cas.cz (I.J.); murafa@iic.cas.cz (N.M.); koci@iic.cas.cz (E.K.); 2Faculty of Mechanical Engineering, JE Purkyně University, Pasteurova 1, 400 96 Ústí nad Labem, Czech Republic; pincz1@seznam.cz (Z.P.); tatjana.brovdyova@ujep.cz (T.B.); stefan.michna@ujep.cz (S.M.); 3Institute of Chemical Process Fundamentals of the Czech Academy of Sciences, Rozvojova 1/135, 165 02 Prague, Czech Republic; fajgar@icpf.cas.cz; 4Institute of Physics of the Czech Academy of Sciences, Na Slovance 1999/2, 182 21 Prague, Czech Republic; alancok@fzu.cz; 5Institute of Mineralogy and Crystallography, Bulgarian Academy of Sciences, Acad. Bonchev 107, 1113 Sofia, Bulgaria; rosica.pn@clmc.bas.bg

**Keywords:** titanium dioxide, microrods, lyophilization, Cu doping, surface structure

## Abstract

This work deals with Cu-modified 1DTiO_2_ microrods (MRs) and their surface properties. The pristine lyophilized precursor Cu_1DTiO_2_, prepared by an environmentally friendly cryo-lyophilization method, was further annealed in the temperature interval from 500 to 950 °C. The microstructure of all samples was characterized by electron microscopy (SEM/EDS and HRTEM/SAED), X-ray powder diffraction (XRD), infrared spectroscopy, simultaneous DTA/TGA thermoanalytical measurement, and mass spectroscopy (MS). Special attention was paid to the surface structure and porosity. The 1D morphology of all annealed samples was preserved, but their surface roughness varied due to anatase-rutile phase transformation and the change of the nanocrystals habits due to nanocavities formation after releasing of confined ice-water. The introduction of 2 wt.% Cu as electronically active second species significantly reduced the direct bandgap of 1DTiO_2_ in comparison with undoped TiO_2_ and the standard Degussa TiO_2__P25. All samples were tested for their UV absorption properties and H_2_ generation by PEC water splitting. We presented a detailed study on the surface characteristics of Cu doped 1DTiO_2_ MRs due to gain a better idea of their photocatalytic activity.

## 1. Introduction

During the past decades, considerable effort has been put into the design and preparation of nanomaterials exhibiting unique physicochemical properties. The main objective of nanoscience is to discover new materials with completely different characteristics at the nanoscale level [1]. The development of new photocatalytic materials with improved stability, low cost, and high photocatalytic activity under solar light are one of the current challenges for photo-electrochemical (PEC) water splitting. Titania, in particular the anatase phase, has been used in PEC [2] very dynamically [3]. It is well established that one-dimensional 1DTiO_2_ (nanorods [4], nanotubes [5], and nanowires [6]) has significant potential in environmental application because of its stability, high specific surface area, and suitable porous structure. However, the large bandgap of TiO_2_ (3.2 eV for anatase), is the main obstacle to obtain highly efficient photocatalysts. The bandgap has a direct impact on the rate of electron-hole (e^−^/h^+^) recombination. Therefore, a huge number of researchers investigate how to narrow the bandgap of TiO_2_ to maximize the utilization of solar energy and to increase the (e^−^/h^+^) pair recombination lifetime, which is a key factor for any TiO_2_-based photocatalyst. One very attractive approach is doping of TiO_2_ with transition metals such as the very low-cost copper (Cu). Cu was previously investigated in this context, including doping of the standard Degussa TiO_2_-P25 [7]. The effect of Cu doping in titania and its importance for photocatalysis has recently been highlighted in several reports [8,9,10,11,12,13,14]. Nevertheless, the results concerning the photocatalytic properties of Cu-doped TiO_2_ nanomaterials are very controversial and this is due to the different methods of preparation used of the materials. Byrne et al. [15] and Obregon et al. [16] indicating the negligible photocatalytic activity of Cu-doped TiO_2_, despite the prevailing anatase phase in samples, arises from the charge recombination at surface lattice defects. Likewise, the recent study of Bensouici et al. has shown the forming of CuO in Cu-doped TiO_2_ thin films decreases its photocatalytic efficiency [17]. Moreover, Wang et al. found that Cu-doped TiO_2_ thin films obtained by RF magnetron sputtering could be used for H_2_ production by water splitting [18]. Also, the incorporation of 0.5 M% of copper ions into the TiO_2_ structure seems to enhance the photoactivity of the system when acidic conditions occur. A possible explanation of this photocatalytic improvement might be related to the stabilization of Cu_2_O species and oxygen vacancies generated in doped TiO_2_ during the preparation procedure using sulfuric acid [9].

Among existing reports, there are no studies on the impact of Cu dopant on the TiO_2_ with 1D morphology. Although many studies focused on the influence of Cu on TiO_2_ microstructure-grain size evolution, specific surface area, and anatase to rutile transformation, scarce experimental research has been carried out in the field of PEC water splitting by Cu-doped 1DTiO_2_ materials.

The Cu^2+^ ionic radius (0.730 Å) is close to the Ti^4+^ radius (0.605 Å) therefore, the dispersion of Cu^2+^ into TiO_2_ crystal lattice is naturally expected by substitutional incorporation of the Ti^4+^ [19]. This substitution with aliovalent Cu^2+^ ion is expected to contribute to the formation of impurity levels, as well as to modify the electronic density of states inside the bandgap of pristine TiO_2_ structure [3].

To the best of the authors’ knowledge, there is no previously published work presenting Cu-doped TiO_2_ microrods (MRs) obtained by annealing of lyophilized 1DTiO_2_ precursor, and their application for H_2_ generation by PEC water splitting. This work presents a complete study of the influence of Cu dopant on the TiO_2_ MRs phase transition from the metastable anatase phase to the stable rutile phase. The characterization of as-prepared materials was carried out using X-ray powder diffraction, Raman, and FTIR spectroscopy, BET/BJH surface analysis. HRTEM microscopy was applied to analyze Cu-doped TiO_2_ MRs grain size evolution at different annealing temperatures. Optical properties and direct bandgap energy (E_g_) measurements were performed using UV-vis spectroscopy. The PEC water splitting and hydrogen evolution were performed in the PEC cell composed of three compartments (for working, counter, and standard saturated Ag/AgCl electrodes).

## 2. Materials and Methods

### 2.1. Materials

Titanium (IV) oxysulfate (titanyl sulfate, TiOSO_4_, Sigma-Aldrich spol. s.r.o., Prague, Czech Republic) was used as a TiO_2_ precursor. Cu(NO_3_)_2_.3H_2_O assay spec. 99%, served as the Cu dopant source. During material synthesis, an aqueous solution of ammonia (NH_3_, hydroxide, purum p.a., 25–29% solution, Fisher Scientific, spol. s.r.o., Pardubice Czech Republic) was used for precipitation of the precursor. AEROXIDE TiO_2_-P25 (Evonik, Prague, Czech Republic) were used for the PEC experiments.

### 2.2. Synthesis of Cu-Doped 1D TiO_2_ MRs

The Cu doped 1DTiO_2_ MRs were prepared via a green freeze-casting method, known as lyophilization (see Scheme 1) [20]. In a typical experiment a mixture of 100 mL of deionized water and 50 mL of crushed ice was prepared in a 350 mL laboratory beaker (step 1). Immediately afterward, 4.8 g of hydrated titanyl sulfate TiOSO_4_ was added to the as-obtained mixture (step 2), and after continuously stirring, 0.118 g of Cu(NO_3_)_2_·3H_2_O was used to yield a calculated value of 2 wt.% of copper to give a blue-colored solution (step 3). Subsequently, the precipitation was carried out at ∼0 °C in aqueous ammonia until the pH reached 8. The precipitate as obtained was transferred into a beaker and resuspended into 350 mL of distilled water. The resulting precursor was immediately lyophilized (freeze-dried) (step 6) at 20 mTorr, at (−54) °C for 72 h by using VirTis Benchtop K lyophilizer, Core Palmer UK, Ipswich, UK). Lyophilized product was isolated in the last step 7. Visualization with SEM microscopy confirmed that the lyophilized precursor labeled Cu/TiO_2__N (Non-annealed) has well-defined 1D MRs morphology. The lyophilized precursor Cu/TiO_2__N was further heat-treated under air at 500, 650, 800, and 950 °C for 1 h in each case (with rate 3 °C min^−1^), and four new samples denoted as Cu/TiO_2__500, Cu/TiO_2__650, Cu/TiO_2__800 and Cu/TiO_2__950 were obtained (Scheme 1).

### 2.3. Methods of Characterization

Prepared materials were characterized by X-ray powder diffraction (XRD), scanning electron microscopy (SEM), transmission electron microscopy (TEM), specific surface area (BET) and porosity (BJH) measurement, thermal analysis (TG/DTA), Fourier transform infrared (FTIR), Raman spectroscopy, UV-Vis spectroscopy and PEC experiments (H_2_ generation by PEC water splitting). The whole set of instrumental characterization is described in our previous publication as you can see in [20,21,22,23,24,25].

Morphology and elemental analysis were investigated using FERA 3 (Tescan, Warrendate, PA, USA) and JSM-6510 LV (JEOL, Peabody, MA, USA) scanning electron microscopes equipped with an energy dispersive X-ray (EDX) detector (Oxford, UK). The measurements were carried out in a low vacuum mode (with a secondary electron detector) at an accelerating voltage of 30 kV. Powdered samples were placed on carbon tape and SEM images were observed without any coating to study their original texture.

Transmission electron microscopy (TEM) was carried out on a JEOL JEM 3010 microscope operated at 300 kV (LaB_6_ cathode, point resolution 1.7 Å) and equipped with an EDS spectrometer. The images were recorded on a CCD camera (Gatan, Pleasanton, CA, USA) with a resolution of 1024 × 1024 pixels. Digital Micrograph and INCA software packages were used for structural and chemical analyses, respectively. Electron diffraction patterns were evaluated using the Process Diffraction program [26].

Room temperature diffraction (XRD) patterns were collected using an X’pert PRO diffractometer (PANalytical, Cambridge, UK) equipped with a conventional X-ray tube (Cu_Kα_ 40 kV, 30 mA, line focus) in transmission mode. An elliptic focusing mirror, a divergence slit 0.5°, a 0.5° anti-scatter slit, and a Soller slit of 0.02 rad were used in the primary beam. A PIXcel (Cambridge, UK) fast-linear position-sensitive detector with an anti-scatter shield and a Soller slit of 0.02 rad was used in the diffracted beam. All patterns were collected in the range of 10 to 80° 2 θ with the step of 0.013° and 400 s per step producing a scan of about 2.4 h. Samples were grounded in an agate mortar in a suspension with cyclohexane. The suspension was then placed on top of a Mylar foil to a transmission sample holder. After solvent evaporation, a thin layer of the prepared sample was then covered with the second Mylar foil. Qualitative analysis was performed with the HighScorePlus software package (PANalytical, version 4.8.0), DiffracPlus software package (Bruker AXS, Karlsruhe, Germany, version 8.0), and the JCPDS PDF-4 database. International Centre for Diffraction Data (JCPDS PDF-4 Database, release 2019/1 ed; ICDD: Newtown Square, PA, USA, 2019). For quantitative phase analysis, we used DiffracPlus Topas (Bruker AXS, version 4.2) with structural models based on the ICSD database (Fiz Karlsruhe–Leibniz Institute for Information Infrastructure. ICSD Database; FIZ Karlsruhe GmbH: Eggenstein-Leopoldshafen, Karlsruhe, Germany, 2019). This program allows the estimation of the weight fractions of crystalline phases through the Rietveld refinement procedure. The estimation of the size of crystallites was performed based on the Scherrer formula as implemented within the DiffracPlus Topas software.

Thermal analysis (TA/MS) measurements were performed on a SetSys Evolution system (Setaram, Prague, Cyech Republic) coupled with a QMG 700 SuperSonic quadrupole mass spectrometer (Pfeiffer, by Setaram) system. Decomposition of prepared samples was performed in an open alumina crucible in argon (60 mL min^–1^), and the sample mass was approximately 10 mg. The used temperature range was 30–1100 °C with a heating rate of 5 °C min^–1^. Gaseous products were measured as the intensity of individually selected fragments, with the *m*/*z* (mass-to-charge ratios) at 16, 17, 18, 28, 30, 32, 44, 46, and 64. FTIR spectra were measured on a Nexus 670 FTIR spectrometer (Thermo Nicolet, Brno, Czech Republic) in the region 4000–400 cm^–1^ at a resolution of 4 cm^–1^ using KBr pellets. Raman spectra were acquired with an XDR AIY 0900237 Raman microscope (Thermo Scientific, Brno, Czech Republic), 256 two-second scans were accumulated with a laser at 532 nm (0.1 mW or 0.5 mW), 25 μm slit under a 10× objective of a microscope (Olympus, Prague, Czech Republic) in full range with the distinction of 4 cm^–1^. Diffuse reflectance UV-Vis spectra were recorded in the diffuse reflectance mode (*R*) and transformed to a magnitude proportional to the extinction coefficient (*K*) through the Kubelka–Munk function, *F*(*Rα*) [27]. A Lambda 35 spectrometer (PerkinElmer, Waltham, MA, USA) equipped with an RSA-PE-20 integration sphere (Labsphere, North Sutton, NH, USA) using BaSO_4_ as a standard was used. The band-gap energy *E*_bg_ was calculated by the extrapolation of the linear part of equation *λ*_bg_ = 1240/*E*_bg_ (eV) [28,29].

### 2.4. Photoelectrochemical Experiments

The water splitting and hydrogen evolution were performed in a photoelectrochemical cell composed of three compartments (for working, counter, and standard saturated Ag/AgCl electrodes) filled with an electrolyte and separated by semi-permeable membranes made of sintered glass. Sulfuric acid (H_2_SO_4_, 0.5 M in distilled water) was used as the electrolyte. As a source of the incident light, the 100W solar simulator (LCS-100, Oriel, Edmonton, AB, USA) was used. The incident power of the light was measured using a light meter (DT-8809A, CEM, Prague, Czech Republic) and the intensity was 710.10^3^ lux, which is equal to 0,104W/cm^2^ at 555 nm. The electrochemical response of layers on incident light during cyclic voltammetry was studied by a potentiostat (2450 SourceMeter, Keithley, Houten, The Netherlands). The aperture between the light source and the electrodes allowed light/dark operation. The material studied was deposited on a fluorine-doped tin oxide-coated glass (FTO-glass) electrode with a working area of about 3 cm^2^. The samples were deposited from an ultrasound dispersed powder material (0.01 g) in 50 mL distilled water onto the electrode (12 mm × 42 mm), immersed in the dispersion overnight. Subsequently, the sample was annealed for 1 h in the air at a temperature of 500 °C. Cyclic voltammetry (CV) data was measured by applying a potential between −0.7 and 0.7 V on the working electrode against a standard saturated Ag/AgCl electrode with a constant scanning speed of 25 mV/s. Four cycles were monitored to evaluate the stability of the photoperiods, reproducibility, and stability of deposited layers [21]. The measurements were repeated three times to evaluate the stability of the studied layers.

## 3. Results and Discussion

This section presents the results obtained during the characterization of the Cu/TiO_2__N and Cu/TiO_2__500, Cu/TiO_2__650, Cu/TiO_2__800, and Cu/TiO_2__950 materials and their interpretation.

### 3.1. X-ray Diffraction (XRD) Characterization

XRD patterns of Cu/TiO_2__N and post-annealed materials (Cu/TiO_2__500, Cu/TiO_2__650, Cu/TiO_2__800, and Cu/TiO_2__950) are shown in Figure 1. Analysis of lyophilized precursor Cu/TiO_2__N confirmed non-crystalline, completely amorphous material. Sample Cu/TiO_2__500 (Figure 1) shown a diffraction pattern consistent with a single phase of TiO_2_ that can be assigned to tetragonal anatase (JCPDS database PDF file 21-1272), indicating that the crystal structure was not changed after Cu doping. When annealed at 650 °C. Rutile polymorph of TiO_2_ (3.7 wt.%) began to appear (JCPDS database PDF 21-1276) (Figure 1 and Figure 2). No diffraction lines of any copper-containing phases were observed in the samples Cu/TiO_2__500 and Cu/TiO_2__650 implying that Cu^2+^ can preferentially substitute Ti^4+^ in the anatase phase before the transition to rutile. A complete transformation of anatase to rutile is achieved at 800 °C (Figure 1). Sample Cu/TiO_2__800 is a mixture of 98.3% rutile and 1.7% CuO (Figure 2). The transformation from thermodynamically metastable anatase to stable rutile is accompanied by segregation of monoclinic copper (II) oxide CuO, known as tenorite (JCPDS database PDF 48-1548). At this temperature, the Cu^2+^ starts to escape from the anatase lattice. Annealing at the highest temperature of 950 °C causes the CuO to grow rapidly, and the amount of CuO increases at the expense of rutile (Figure 1). The percentage estimation (wt.% of phases) of any individual phase in each stage of annealing temperature, based on the Rietveld refinement, is shown in Figure 2. 

The phase composition, lattice parameters, and crystal sizes obtained by Rietveld refinement are given in Table 1. The changes in lattice parameters as a function of temperature shown that in samples Cu/TiO_2__500 and Cu/TiO_2__650, the lattice parameter *a* decreases, and the parameter *c* increases with the increasing temperature. The increase of *c* axis lattice parameter with metal ions doping (Mo and Sc) in anatase NCs has already been reported in our recent works [22,23,24,25]. This further supports the substitutional incorporation of Cu^2+^ ions into the anatase lattice. Also, the ionic radii of Cu^2+^ ions are higher than that of Ti^4+^ (R_Cu2+_ = 0.730 nm vs. R_Ti4+_ = 0.605 nm) and such changes of lattice parameters and lattice volume are expected [30]. We should recall, that beyond 650 °C, the anatase starts to transform to rutile, and lattice parameters in samples Cu/TiO_2__800 and Cu/TiO_2__950 are estimated for rutile. It is observed that in both samples, parameter *a* is almost invariant with temperature, whereas the parameter *c* slightly decreases. The Cu doping progressively reduces the concentration of Ti in the Cu/TiO_2__800 and Cu/TiO_2__950 samples. Also, a change of the cation distribution could be expected since the arising of the tenorite phase.

The average crystallite size of Cu/TiO_2__500, Cu/TiO_2__650, Cu/TiO_2__800, and Cu/TiO_2__950 samples has been estimated from the XRD data using Scherrer’s formula [31]. The values presented in Table 1, indicated that the grain growth of anatase and rutile varies exponentially with annealing temperature. Once formed at 650 °C, the rutile NPs grew very rapidly and coarsening up to 157 nm at 950 °C (sample Cu/TiO_2__950) [32]. At this temperature, an energetically stable TiO_2_ polymorph is formed. We believed that substitutional Cu^2+^ doping plays an important role in anatase to rutile phase transformation and NPs growth. Here, the transformation has occurred at a higher temperature than that reported in the literature (400 °C for undoped anatase [33].

### 3.2. Effect of Cu and Temperature on Surface Area (BET) and Porosity (BJH)

The BET/BJH analysis was performed to examine the surface area and porosity of Cu/TiO_2__500, Cu/TiO_2__650, Cu/TiO_2__800, and Cu/TiO_2__950 samples. Table 1 shows that increasing temperature led to a decrease in the surface area due to NPs growth. Sample Cu/TiO_2__500 with 100% anatase structure and smaller NP size shown the highest surface area, followed by sample Cu/TiO_2__650 (96.3% anatase and 3.76% rutile). During annealing at 800 and 950 °C, both samples Cu/TiO_2__800 and Cu/TiO_2__950 with almost pure rutile structure (Figure 2), showing a tendency for decreasing the specific surface area. An increasing the degree of agglomeration by the effect of coarsening occurring in these materials. The specific surface area of sample Cu/TiO_2__950 decreases to 2.7061 m^2^g^−1^ suggested by the largest size of rutile NPs and their faster growth compared to anatase. According to the IUPAC nomenclature, materials with a pore size of 2–50 nm, are referring to the mesoporous compounds. Therefore, annealing of lyophilized Cu/TiO_2__N precursor has led to the formation of mesoporous Cu/TiO_2__500, Cu/TiO_2__650, Cu/TiO_2__800, and Cu/TiO_2__950 materials. While surface area decreases, the pore radius of Cu/TiO_2__500, Cu/TiO_2__650, Cu/TiO_2__800, and Cu/TiO_2__950 samples increase probably due to elimination of chemically bonded water during annealing up to 950 °C (see Table 1). These results are supported by TA/MS analysis presented in the Appendix A.

### 3.3. Scanning Electron Microscopy (SEM) and Energy-Dispersive X-ray Analysis (EDS)

The morphology and surface roughness changes in Cu/TiO_2__500, Cu/TiO_2__650, Cu/TiO_2__800 and Cu/TiO_2__950 materials with annealing temperature are presented in a set of SEM micrographs.

#### 3.3.1. SEM/EDS of Sample Cu/TiO_2__500

Figure 3a is an SEM image of the Cu/TiO_2__500 sample at low magnification confirmed its 1D morphology. MRs with 10–20 μm in length and 1 μm in diameter are visible. Figure 3b, which is a high magnification of the red boxed area in Figure 3a documents that the 1D Cu/TiO_2__500 MRs are composed of densely stacked thin layers with an outermost layer keeping a very clean smooth surface.

#### 3.3.2. SEM/EDS of Sample Cu/TiO_2__650

In Figure 4a,b are shown SEM images of a Cu/TiO_2__650 sample at low and high magnifications, respectively. The layered morphology of the MRs and their length were preserved at annealing at 650 °C.

#### 3.3.3. SEM/EDS of Sample Cu/TiO_2__800

While annealing at 800 °C, the MRs morphology was preserved (Figure 5a), but the surface of the Cu/TiO_2__800 material was changed significantly. Figure 5b,c presented the formation of spherical cavities with a size of 20–50 nm in diameter on the outermost layer attributed to water, NO_x,_ CO_x_, and SO_x_ species releasing upon annealing (see Appendix A). Furthermore, newly wave-like bulge features can be observed in Figure 5c,d. We can suggest that changes in surface morphology can be provoked by anatase to rutile transformation and tenorite (CuO) NCs segregation on the MRs surface (Figure 5d).

#### 3.3.4. SEM/EDS of Sample Cu/TiO_2__950

It is evident that Cu/TiO_2__950 MRs showed different shapes from Cu/TiO_2__500, Cu/TiO_2__650, Cu/TiO_2__800 MRs; they are constructed from 1D oriented close-packed single crystals. One can observe that the Cu/TiO_2__950 MRs are high-temperature stable structures (Figure 6a,b) preserved 1D morphology even at such high temperatures.

EDS analysis (Table 2) confirmed the presence of oxygen, titanium, and copper with average values in wt.% and at.%. It can be noted that the stoichiometric ratio of titanium to oxygen in all annealed samples differs from the stoichiometric ratio of titanium dioxide (TiO_2_). The substitution of Cu^2+^ into the Ti^4+^ site in anatase could generate oxygen vacancies (V_o_) and titanium interstitials (Ti_i_) due to charge balance. Also, the crystallization of CuO NCs at higher temperature could lead to significant non-stoichiometry in Cu/TiO_2__500, Cu/TiO_2__650, Cu/TiO_2__800 and Cu/TiO_2__950 materials.

### 3.4. High-Resolution Transmission Electron Microscopy (HRTEM) and Selected Area Electron Diffraction (SAED)

An HRTEM and SAED study was applied to verify the structures of Cu/TiO_2__500, Cu/TiO_2__650, Cu/TiO_2__800, and Cu/TiO_2__950 materials on an atomic level.

#### 3.4.1. HRTEM/SAED of Sample Cu/TiO_2__500

A bright filed image of Cu/TiO_2__500 MRs taken by HRTEM demonstrated 1D morphology (Figure 7a). The corresponding SAED pattern (Figure 7b) exhibited concentric diffraction rings with intermittent dots, implying that the Cu/TiO_2__500 is polycrystalline material. The analysis of a ring pattern identified anatase TiO_2_ (JCPDS database PDF file 21-1272) with crystal planes (101), (004), and (200) and relevant interplanar spacing of 0.352 nm, 0.237 nm, and 0.189 nm. The indexing of the SAED pattern revealed no Cu containing NCs or nanoclusters, suggesting that Cu is incorporated in anatase lattice. These results corroborated with XRD analysis.

#### 3.4.2. HRTEM/SAED of Sample Cu/TiO_2__650

Low magnification HRTEM image and SAED pattern of the Cu/TiO_2__650 MRs is shown in Figure 8a,b. According to the SAED pattern, it is an anatase phase of titanium dioxide (JCPDS database PDF file 21-1272) with an identical structure as the Cu/TiO_2__500 material. At higher magnification (Figure 8c,d) it is possible to observe well-crystallized material formed by the spontaneous agglomeration of numerous closely packed anatase NCs with a size of 10–15 nm into 1D oriented MRs. Even XRD analysis confirmed 3.73 wt.% of rutile NCs in Cu/TiO_2__650 material, we did not observe rutile diffraction spots by indexing the Cu/TiO_2__650 SAED pattern.

#### 3.4.3. HRTEM/SAED of Sample Cu/TiO_2__800

Low magnification HRTEM image and SAED pattern of the Cu/TiO_2__800 material are shown in Figure 9a,b. One can see elongated close-packed NCs and spot SEAD pattern confirmed that annealing at 800 °C led to larger and still 1D oriented NCs. Here, we indexed crystal planes (110), corresponding to the interplanar spacing of 0.324 nm of rutile TiO_2_ (JCPDS database PDF file 21-1276). The HRTEM revealed that there are many nanocavities inside the NCs building the MRs, visible at higher magnification (Figure 9c). Nanocavities appeared as white spots at the surface of Cu/TiO_2__800 MRs. Similar findings on the nanocavities detected by HRTEM were reported by [34,35]. Additionally, here we can detect not only spherical-shaped nanocavities (see SEM section), but also hexagon-shaped nanocavities (Figure 9c,d), which could be formed due to the six-fold symmetry in ice crystals grown/melting from water vapor upon lyophilization/annealing processes.

#### 3.4.4. HRTEM/SAED of Sample Cu/TiO_2__950

It is evident from the HRTEM image (Figure 10a) that Cu/TiO_2__950 MRs show 1D morphology, but with different nanocavity sizes and distribution compared with Cu/TiO_2__800. The higher annealing temperature generated larger nanocavities at the surface that can be explained by the enhanced coarsening of ice crystals in Cu/TiO_2__950 due to freezing–thaw cycling and annealing. The interpretation normally set up forward the dissolution–regrowth mechanism: first, the smallest ice particles are dissolved upon heating, leading to fewer nuclei. Second, upon lyophilization, the amount of dissolved ice crystals is regrown on the remaining nuclei, leading to larger particles. The larger ice particle becomes soft and liquid with increasing temperature up to 950 °C, and further, evaporate leaving behind bigger cavities on the surface [36]. Furthermore, one can observe a small dark spot on the surface of Cu/TiO_2__950 MRs (marked with the red arrow in Figure 10a). We can speculate that this is CuO crystallized during annealing and segregated on the surface of rutile grains. The precipitation of tenorite (CuO) on the surface of Cu/TiO_2__950 MRs can be explained as a change of an inhomogeneous structure over the temperature in TiO_2_-CuO solid solution. The process involves matter relocation, i.e., crystallization of small CuO NCs on the surface of larger TiO_2_ NCs. This process can be explained based on the Oswald ripening phenomenon [37]. The IUPAC in 2007 recommended the definition of Ostwald ripening as the “dissolution of small crystals or soil particles and the redeposition of the dissolved species on the surfaces of larger crystals or soil particles.” [38]. The ripening process occurs in Cu/TiO_2__950 MRs because larger particles TiO_2_ NPs become spatially quite close to each other which leads to the coarsening and rise of their crystallite’s size during annealing (Table 1). Also, such a process is more energetically favored since giving rise to an apparent higher solubility for the smaller NCs to segregate on the surface of the biggest one [39].

SEAD of the Cu/TiO_2__950 MRs (Figure 10b) revealed a spot diffraction pattern. It should be pointed out that the presence of strong and well-defined spots is due to the increase in rutile grain size to the dimensions of 300 × 450 nm single crystals. The structure of rutile NC (Figure 10c) taken from the red boxed area in Figure 10a, fitted very well the tabulated rutile (JCPDS database PDF file 21-1276). Also, a magnified part (inset in Figure 10d) of the red marked region, revealing the atomic resolution image with nanooctahedra atomic arrangement, which is characteristic of stable rutile TiO_2_ structure. The HRTEM results of Cu/TiO_2__950 are consistent with XRD analysis. The mean grain size in Cu/TiO_2__950 material is closely related to the amount of rutile phase; the more rutile percentage found in Cu/TiO_2__950, the bigger the mean grain size was estimated (Table 1, Figure 10a).

### 3.5. Raman Spectroscopy

As revealed by XRD, the Cu/TiO_2__N is highly amorphous. The Raman spectrum of the sample shows broad convolution bands, typical for highly unoriented titania (Figure 11). The bands, marked by an asterisk were recognized in the TiO_2_(B) structures [40]. In the samples, annealed at 500 and 650 °C, five bands representing anatase vibration modes (3E_g_ + 2B1_g_ + A1_g_) appear and confirm the XRD results. In the high-temperature samples (800 and 950 °C), the rutile structure has been proved by bands at 609, 444, and 243 cm^−1^. On the other hand, tenorite CuO observed as weak reflections in the XRD are not visible at expected 295 and 345 cm^−1^ due to low concentration of copper (Figure 11).

### 3.6. Fourier-Transform Infrared Spectroscopy

Figure 12 shows the infrared spectra of the examined samples annealed at temperatures of 500, 650, 800, and 950 °C, and nonannealed Cu/TiO_2__N also. In the area above the wavenumber of 3000 cm^−1^, a wide absorption band with maxima at 3385 cm^−1^ is visible, which corresponds to the stretching vibration of water-bound on the surface of substances. Following the stretching vibration, the samples show a bending vibration at 1632 cm^−1^. The band visible at 3170 cm^−1^ belongs to the stretching vibration of the ammonium cation and the value of 1400 cm^−1^ belongs to its bending vibration, which is typical of ammonium salts. The presence of ammonium ions is related to the use of aqueous ammonia solution during the preparation of the default sample. The band position is also typical of nitrate anion, having weaker bands at 1050 and 830 cm^−1^, visible as shoulders in the as-prepared sample. Annealing of the Cu/TiO_2__N leads to the disappearance of these bands due to the decomposition of ammonium nitrate. In the as-prepared sample, the formation of Ti–O, Ti–OH, and Ti–O–Ti bonds is demonstrated by a broadband at about 555 cm^−1^. In the annealed Cu/TiO_2__500, Cu/TiO_2__650, Cu/TiO_2__800, and Cu/TiO_2__950 MRs, two bands corresponding to Ti–O and Ti–O–Ti appear (620 and 470 cm^−1^ in the anatase structures, 660 and 500 cm^−1^ in the rutile structures).

### 3.7. Optical Spectroscopy Results

#### 3.7.1. UV-Visible Analysis

UV-visible studies and direct bandgap estimation of the Cu/TiO_2__N, Cu/TiO_2__500, Cu/TiO_2__650, Cu/TiO_2__800 and Cu/TiO_2__950 materials are shown in Figure 13a–h. Also, the results of the pristine 1D TiO_2_ MRs prepared by the same method and standard TiO_2_-P25 are presented for comparison.

All annealed materials show a clear red shift and absorption in the visible region. By increasing the temperature, considerable bandgap narrowing is observed as depicted in Figure 13e–h. The estimated band gap of pristine TiO_2_ was found to be 3.50 eV, whereas the band-gap of Cu/TiO_2__500, Cu/TiO_2__650, Cu/TiO_2__800 and Cu/TiO_2__950 is 3.24 eV, 3.21 eV, 3.06 and 3.06 eV, respectively. One plausible explanation for the bandgap narrowing could be the presence of Ti_i_ defects in TiO_2_ NCs [41]. It is reported that the formation of Ti_i_ is generally due to the preparation conditions and annealing under air [42]. Doping with 2 at.% Cu atom substitutes some of the Ti sites in the TiO_2_ lattice, which could lead to a strong d–p coupling between Cu and O, and to O 2p orbital upward movement, resulting in a reduction of the bandgap. Further, Cu 3d orbital could form an impurity band above the TiO_2_ valence band causes a reduction in the overall bandgap [43]. Since annealing at a temperature higher than that of 650 °C promoted the rutile polymorph formation, there is a high chance of having a defect contributed to the E_g_ as a result of structural transformation. Theoretical studies on metal-doped TiO_2_ predict also the formation of an empty band very close to the conduction band due to high vacancy concentrations [44]. From the absorption spectra, it is evident that the absorption edge of the TiO_2__P25 material is near 347 nm, whereas the redshift associated with the Cu doping in Cu/TiO_2__500 and Cu/TiO_2__650 MRs may arise as a result of surface trap centers generated by V_o_ defect sites after Cu doping. Our results obtained by EDS analysis are in line with this observation. Recall, that Cu/TiO_2__800 and Cu/TiO_2__950 materials are mixtures with 98 wt.% of rutile and a small amount of CuO (see Table 1). To our knowledge, there are no publications have been reported on the optical properties of photocatalysts with such composition and structure. Both materials have shown a bandgap of 3.06 eV, which can be related to the different strategies of how the conduction-valence band edge changed from anatase to rutile structure. It is worth mentioning that CuO is a well-known p-type semiconductor owing to suitable direct bandgap energy (2.17 eV) and an optical gap (2.62 eV) predicting its application as a promising photocatalyst for visible light [43]. However, the photocatalytic efficiency of CuO has been low because of its even-parity symmetry of the conduction and valence band minimum states that prohibit the band edge radiative transition [45,46].

#### 3.7.2. PEC Water Splitting

Photocatalytic activity of the synthesized Cu/TiO_2__500, Cu/TiO_2__650, Cu/TiO_2__800, and Cu/TiO_2__950 MRs was s quantitatively examined for H_2_ generation by PEC water splitting. Figure 14 shows the effect of Cu doping on photocatalytic activity under solar light irradiation. It is evident, that even a significant reduction of bandgaps, the Cu/TiO_2__500, Cu/TiO_2__650, Cu/TiO_2__800, and Cu/TiO_2__950 MRs did not satisfy the criteria for efficient photocatalysts and show weak H_2_ generation from H_2_O splitting.

PEC experiments were used to evaluate both the electrochemical (in the dark) and photoelectrochemical (under light irradiation) behavior of the prepared layers on FTO electrodes under the same experimental conditions. During the PEC experiment, the sample stabilization test was first performed, which is consisted of charging/discharging it in the dark This was followed by the experiment itself, where the sample was illuminated by a lamp that simulated the spectrum of the solar radiation with intensity 0.104 W/cm^2^. A dark test was then also performed. The photoactivity of the sample is then determined by comparing the test in the dark and under solar light. The voltammograms are shown in Figure 14, where two photoelectrochemical curves, namely the 2nd and 7th cycle are depicted. Higher cycles (8th–12th) show similar curves which means that system is relatively stable. The dark electrochemical curves do not show major differences and therefore only one typical curve is demonstrated.

The electrochemical curves are dominated by a peak at −0.51 V vs. Ag/AgCl reference electrode (−0.360 V against RHE), which is typical for oxidation Cu^0^ → Cu^+^ in the oxidation part of the curves. The reduction parts of the curves show broad reduction peaks between −0.4 and −0.6 V. The minima of these reduction peaks are shifted to the higher potential in comparison with the oxidation peak, which means that the oxidation/reduction process is not fully reversible. The PEC experiments were repeated 12 times and the curves were observed to shift due to oxidation/reduction processes in the samples. Under −0.6 V, evolving of gaseous hydrogen was observed on the platinum counter electrode. The highest activity was observed with the sample annealed at 650 °C (Cu/TiO_2__650). In the corresponding voltammograms we observe the highest difference between dark and light curves, reaching about 1 mA. In the Cu/TiO_2__650 sample, the lowest degree of Cu oxidation/reduction is proved by the weakest corresponding Cu/Cu^+^ peaks. In this sample, the photoelectrochemical activity is even improving with several cycles, while other samples show slowly decreasing photoactivity between 7th and 12th cycles. With increasing voltage, the current approaches zero, and voltages above +0.8 V induce releasing particles from the FTO substrates and damaging of the layers. Preparation of highly adhesive doped TiO_2_ films can solve loose binding between the substrate surface and TiO_2_. Due to this effect, there is not possible to achieve a higher potential to reversibly reduce copper oxides. The photoactivity of samples Cu/TiO_2__500 and Cu/TiO_2__800 is much weaker in comparison with Cu/TiO_2__650 and it is gradually decreasing with the rising number of the PEC cycles. Except for non-reversible oxidation of Cu, low activity of the samples can be explained by (a) the rapid recombination of e^−^ and h^+^ due to the occupancy of active surface sites by the Cu atoms, (b) the surface nanocavities acting as point defects, which interrupt charge carrier transportation (instead of providing multiple reflections of the adsorbed light) and (c) photocorrosion suffering when materials are in contact with water (although that Cu doping manipulated the electronic structure of Cu/TiO_2__500, Cu/TiO_2__650, Cu/TiO_2__800 and Cu/TiO_2__950 materials). In recently published articles [47,48] the authors discussed how CuO based materials can be utilized for PEC hydrogen evolution: the reduction potential of p-type low bandgap CuO in aqueous electrolyte lies above the reduction potential of wateFr. This implies the possibility of a competitive reduction of water and the photoabsorber itself, leading to the formation of reduced copper species and metallic copper, and consequently to a significant change of the morphology of the photoabsorber due to photocorrosion. The abovementioned study reveals that the transformations taking place in copper (II) oxide electrodes, with special attention to the role of photo-induced electrons, are the most important points, which could be considered for a various copper amount (% of the Cu dopant). Yoong [7] reported that Cu dopant concentration and size range of Cu NPs (20 and 110 nm) are the main factors affecting the properties of Cu-doped TiO_2_ photocatalysts for hydrogen production under UV-Vis light irradiation. Among the dopant loading ranging from 2–15 wt.% on Degussa P25 TiO_2_, and annealing from 300 to 500 °C, it was observed that sample 10 wt.% Cu/TiO_2_/300 °C yielded the maximum hydrogen generation. In contrast to this work, Bandara et al [49] reported a copper loading of 7 wt.% into TiO_2_ to be the best photocatalyst for the reduction of H_2_O under sacrificial conditions (CH_3_OH scavenger for photoinduced holes). The amount of CuO and its crystalline structure were found to be crucial for the catalytic activity since the negative shift in the Fermi level was caused by an accumulation of excess electrons in CuO. It is worth mentioning that the concentration of Cu dopant strongly influenced the oxidation state of Cu species (Cu^0^, Cu_2_O, CuO), the rate of rutilization process, and photocatalytic properties of the Cu-TiO_2_ system. A report by Song et al. [50] documented that the evolution of Cu^0^ toward Cu^1+^ and Cu^2+^ progressively lead to a lowering in photocatalytic activity. Colon et al. [51] observed that 0.5 at.% Cu loading and annealing up to 600 °C led to the highest photocatalytic activity due to prevailing anatase phase, high surface area, and copper ions in form of Cu^1+^ as photoactive homogeneously dispersed Cu_2_O species into the anatase matrix. Stabilized Cu_2_O NPs are responsible for the improved electron transfer to the surface oxygen and further photocatalytic activity. According to XRD analysis, TiO_2_ systems with 1 at.% Cu showed lower anatase content; when Cu concentration is higher than that of 0.5 at.% then accelerated rutile microdomains are observed. Also, 1 at.% Cu content can generate CuO species; the existence of Cu^2+^ was considered as electron/hole recombination center, which progressively may deteriorate the photoactivity. Therefore, careful Cu amount loading and controlled Cu^2+^/Cu^1+^ redox reactions can adjust the satellite CuO/Cu_2_O structures, which participated in TiO_2_ photoreactivity [9].

We could suggest that 2 at.% Cu doping is not the optimum amount in TiO_2_ lattice to produce enhanced H_2_ generation for PEC water splitting. Modification of Cu dopant concentration, and involving of appropriate scavenger for photoinduced holes, maybe a more promising strategy for utilization of Cu-modified 1DTiO_2_ materials in PEC water splitting processes. 

## 4. Conclusions

We have developed an original and green lyophilization based method for the synthesis of anatase TiO_2_ MRs doped with 2 wt.% Cu. We used a nitrogen-containing Cu(NO_3_)_2_ reagent as a source of Cu, reported to affect the photocatalytic activity of TiO_2_ [52]. A series of new materials with MRs morphology denoted as Cu/TiO_2__500, Cu/TiO_2__650, Cu/TiO_2__800, and Cu/TiO_2__950 were obtained after annealing at 500, 650, 800, and 950 °C from lyophilized 1DTiO_2_ MR precursor. We have found that the photocatalytic activity of Cu/TiO_2__500, Cu/TiO_2__650, Cu/TiO_2__800, and Cu/TiO_2__950 MRs is strongly dependent on their phase structure, crystallite size, specific surface areas, and pore structure. The substitution of 2 at.% Ti^4+^ with Cu^2+^ ions has a substantial influence on the materials’ optical and electronic properties. The optical bandgap of Cu/TiO_2__500, Cu/TiO_2__650, Cu/TiO_2__800 and Cu/TiO_2__950 MRs is reduced significantly. All materials show a clear red shift and absorption in the visible region. Although Cu doping modified the electronic structure of the Cu/TiO_2__500, Cu/TiO_2__650, Cu/TiO_2__800, and Cu/TiO_2__950 materials, all of them appeared slightly active during the PEC water splitting experiments. The PEC water splitting activity is decreasing in the order: Cu/TiO_2__650 >> Cu/TiO_2__500 > Cu/TiO_2__800 >> Cu/TiO_2__950. The low activity can be explained by (a) the rapid recombination of e^−^ and h^+^ due to the occupancy of active surface sites by the Cu atoms, deteriorating the chance of photoexcited e^−^ and h^+^ to participate in surface chemical reactions, (b) photocorrosion suffering due to contact with water [53] (c) nanocavities acting as recombination charge centers instead to confine light into their volume and (d) negligible specific surface area S_BET_ (reduced from 10.8 to 2.7 m^2^g^−1^) with the rise of temperature, even preserved mesoporous structure.

We presented a detailed study on the surface properties of Cu-doped 1DTiO_2_ MRs to provide a better understanding of their chemical and physical properties and photocatalytic activity. In the future, our attention will be focused on the synthetic conditions for producing anatase doping with less than 2 wt.% Cu. Some computational works can also be provided due to predict the preparation conditions leading to an efficient photocatalyst, combining n-type TiO_2_ and p-type CuO, with eventual n-p junction conductivity.

## Data Availability

Data is contained within the article or Appendix A.

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
