# Peer review of "Surface Properties of 1DTiO2 Microrods Modified with Copper (Cu) and Nanocavities"

_nanomaterials, 2021, doi:10.3390/nano11020324_

Round 1

Reviewer 1 Report

This work deals with Cu modified TiO2 microrods (MRs) and their surface properties. The samples are well studied X-ray powder diffraction, scanning and transmission electron microscopy, infrared spectroscopy, thermal analysis, and surface – porosity measurements.

In this work the authors present a detailed study on the surface properties of Cu doped 1DTiO2 MRs to provide a better understanding of their chemical and physical surface properties and photocatalytic activity.

I kindly ask the authors not to use the term "chapter". We are not writing a book or a dissertation, so they can use the term "section".

I noticed the authors studied the phase composition of their samples, calculated the lattice parameters, and crystal sizes obtained by Rietveld refinement. Which software have they used?
Please mention the software, and cite it in a reference.

Have the authors coated their samples with Au or Au/Pd prior to SEM? Please include this information in the experimental section.

The authors study the photocatalytic activity of the synthesized Cu/TiO2 MRs, quantitatively examined for H2 generation by PEC water splitting.

a) please mention the solar light simulator you used, and the incident power in W/cm2

b) it is quite critical to check the repeatability of the samples. I kindly ask the authors to re-use their samples at least for 3 times for H2 generation by PEC water splitting.

A few typos should be corrected.

I suggest the manuscript entitled "Surface properties of 1DTiO2 microrods modified with copper (Cu) and nanocavities" to be accepted for publication in MDPI Nanomaterials after the above minor revision.

Author Response

The authors are thankful to reviewer No.1 (#R1).

The reviewer’s’ comments have definitely helped us to improve the quality of the manuscript.

  1. I kindly ask the authors not to use the term "chapter". We are not writing a book or a dissertation, so they can use the term "section".

The reviewer is right that our statement is not clear. We provided changes according to the #R1 suggestion (see line 204, please).

  1. I noticed the authors studied the phase composition of their samples, calculated the lattice parameters, and crystal sizes obtained by Rietveld refinement. Which software have they used?
    Please mention the software, and cite it in a reference.

Information of software using for Rietveld refinement is mentioned in the part 2.3 Methods of characterization, line 162 (DiffracPlus Topas software)

  1. Have the authors coated their samples with Au or Au/Pd prior to SEM? Please include this information in the experimental section.

Information for the SEM experiment is provided in part 2.3 Methods of characterization, lines 134-136 (samples were not coated, their original texture was studied).

  1. The authors study the photocatalytic activity of the synthesized Cu/TiO2 MRs, quantitatively examined for H2 generation by PEC water splitting.
  2. a) please mention the solar light simulator you used and the incident power in W/cm2

We add information about solar light simulation in section 2.4 Photoelectrochemical experiments, see lines 185-186, please.

  1. b) it is quite critical to check the repeatability of the samples. I kindly ask the authors to re-use their samples at least 3 times for H2 generation by PEC water splitting.

We thank the reviewer for such a great idea to check the repeatability of the samples under simulated solar light. We appreciate the significance of this suggestion since we collected more useful information about their optical and stability properties. Updated results are presented in section 3.7.2 PEC water splitting. Check to please the text and Fig.14.

  1. A few typos should be corrected.

English grammar and spelling were checked by a professional editor.

Reviewer 2 Report

The data and results in this paper are well presented overall.  The motivation of work well explained with sufficient background.

The degree of characterization is extraordinary. However, it would be good, if possible, to state how repeatable the synthesis is in terms of the morphology features of samples, if known. 

The research design would have been better if one can compare more variation of sample composition even if means less characterization. It seems that is planned for future work. 

In Results of PEC section, the results (which appear disappointing) show no photoactivity. These results sound be compared to others who also looked at Cu doped TiO2 as it appears some have shown photoactivity as others did not and presumably there are differences in samples. A comparison of sample attributes of Cu:TiO2 would be much more informative than discussion of an entirely different system of CuO - especially since earlier results are "controversial". Also in this section the cyclic voltammetry figures should be formatted better.

Some other significant edits to do are:

explain what M% is

the values of ionic radii of Cu2+ and Ti4+ are not consistent in paper.

NH4OH and Fisher Scientific is misspelled in paper.

In Fig 1 caption, N,A,R, T should be defined. Also, better to define anatase consistently as anatase. One occurrence refers to it as tetragonal anatase. (line 207)

Line 213 a statement how % CuO was determined should be stated.

Fig 2, symbol legend seems to be missing letters.

Line 232, word confirm is too strong. Having consistency with other work supports not confirms.

The pore size in Table 1 should be average I suspect.

Stable is misspelled in line 351.

Fig 13 a caption should without "compared"

In supplemental figure it be good to label respective masses observed as to what species it corresponds to. In in mass loss, it be good to reference the compounds evolved not the mass fragments observed.

Author Response

The authors are thankful to reviewer No.2 (#R2).

The reviewer’s’ comments have definitely helped us to improve the quality of the manuscript.

  1. The degree of characterization is extraordinary. However, it would be good, if possible, to state how repeatable the synthesis is in terms of the morphology features of samples if known. 

Thank you #R2 for your reasonable question. The microrods morphology of 1D TiO2 is well repeatable. We yet published the results for undoped 1D TiO2 (see Ref. 25). Also, metal doping did not change the morphology because of the use of TiOSO4 precursors with preserved 1D morphology. Check to please our recent group´s publication Metatitanic acid pseudomorphs after titanyl sulfates: Nanostructured sorbents and precursors for crystalline titania with desired particle size and shape, Crystal Growth and Design, 17 (2017) 6762-6769.

  1. The research design would have been better if one can compare more variation of sample composition even if means less characterization. It seems that is planned for future work. 

Currently, we are preparing Ag and Au doped TiO2; the SEM and TEM confirmed again 1D MRs morphology.

  1. In the Results of PEC section, the results (which appear disappointing) show no photoactivity. These results sound be compared to others who also looked at Cu doped TiO2 as it appears some have shown photoactivity as others did not and presumably there are differences in samples. A comparison of sample attributes of Cu: TiO2 would be much more informative than a discussion of an entirely different system of CuO - especially since earlier results are "controversial". Also in this section, the cyclic voltammetry figures should be formatted better.

We thank the reviewer for such a great idea to provide more relevant discussion focused on the Cu-TiO2 system only. We appreciate the significance of this suggestion and updated the discussion by comparing our results and yet published outcomes from other authors. New references in this context were used and added to the list of references.

Yoong et al Development of copper-doped TiO2 photocatalyst for hydrogen production under visible light, Energy, 34, 2009, 1652-1661

Bandara et al. Highly stable CuO incorporated TiO2catalyst for photocatalytic hydrogen production from H2O, J Photochem. Photobiol. Sci., 2005, 4, 857K.Y. Song et al. Photocatalytic activity of Cu/TiO2 with the oxidation state of surface- loaded copper, Bull. Korean Chem. Soc., 20 (1999), 957G. Colon, Towards the hydrogen production by photocatalysis, Appl. Catal. A: General, 518, 2016, 48.G. Colon et al. Cu-doped TiO2 system with improved photocatalytic activity, Appl Catal Environmental, 67, 2006, 41 

Also, we updated Fig. 14 presented the voltammetry results

  1. Some other significant edits to do are:

explain what M% is – M is mass Molarity. Molar concentration (mol/L)

the values of ionic radii of Cu2+ and Ti4+ are not consistent in the paper – we have checked the values of ionic radii of Cu and Ti, and did them consistently in the whole manuscript.

NH4OH and Fisher Scientific is misspelled in the paper – we corrected the misspelling

In Fig 1 caption, N,A,R, T should be defined. Also, better to define anatase consistently as anatase. One occurrence refers to it as tetragonal anatase. (line 207)

Line 213 a statement of how % CuO was determined should be stated.

We replaced Fig.1 with a new one, where a better cation was performed. Also, we clarified how % CuO was estimated (see section 3.1 please, Fig. 1 and lines 229-231)

Fig 2, symbol legend seems to be missing letters.

We replaced Fig 2 with a new one with corrected English names of compounds (see section 3.1 please, and Fig.2)

Line 232, word confirm is too strong. Having consistency with other work supports not confirms. – Thank you for this suggestion. We provided changes according to your suggestion,

The pore size in Table 1 should be average I suspect. – Yes. This is the average pore radius. We corrected it in Table 1

Stable is misspelled in line 351.- Thank you! We corrected the spelling.

Fig 13 a caption should without "compared" – Thank you! We updated the caption of Fig.13 according to your suggestion.

In the supplemental figure, it is good to label respective masses observed as to what species it corresponds to. In the mass loss, it is good to reference the compounds evolved not the mass fragments observed. - We thank the reviewer! We appreciate the significance of this suggestion and updated the whole SI file focused on DTA/MS results. We believe that now this part is correct.

Reviewer 3 Report

The paper itself is good, great work carried out, topic is new and important. However, in the present form I would not recommend to publish it.

  1. English should be checked, and besides grammar, terminology should be carefully revised. In figure 2 spelling of anatase, rutile and tenorite is not English. When describing FTIR results, instead of deformation and valence vibrations one should write bending and stretching, and maybe not only.
  2. Presenting the data of measurements one should take into account the accuracy of methods. The specific surface area can hardly be measured with the precision of 1%, and if the instrument shows the value of, say, 2,7061 m2/g, as in Table 1 (5 meaningful figures!), one has to round it up to at least 2.70, or better 2,7. Even the mass of the probe cannot be measured better than +/- 0.1 mg! The same question could be put about lattice parameters in the table, or the data on the composition of samples shown in Table 2. Maybe, some words should be said about the accuracy in the text, if the authors find that last figures in these data are necessary.
  3. The authors used samples doped by 2% of Cu, however it was shown that maximum photocatalytic activity for titania doped with Al, Sc or V is reached at much lower dopant concentrations and then seriously decreases (see, e.g., Murashkina et al, J. Phys. Chem. C, 119(2015), no.44, p.24695 or Murzin et al, Top. Catal., 2020, p.1, and refs therein). Why not at least to discuss possible dependence of activity on the amount of dopant ?

Nevertheless I find the paper worth publishing in Nanomaterials after serious revision, and I wish the authors to finish it in the coming New 2021 year!

Some misprints, certain drawbacks or typical mistakes:

1) The abbreviation MRs first appeared in abstract, line 31, before it is explained in the text (line 78). Why not to do it at first appearance in line 20?

2) line 110:  he lyophilized precursor

3) line 153: This program permits… better  This program allows or enables… 

4) lines 229-230: the lattice parameter a decrease and the parameter c increase with increasing of temperature. better: the lattice parameter a decreases and the parameter c increases with the increasing temperature.

5) line 267:  mesoporous annealing   should be   mesoporous, annealing  without comma no sense;

6) line 270: pore radius of … samples increase  should be  pore radius of … samples increases...

7) lines 554, 556 and not only:   instead typical for… better  typical of

Author Response

The authors are thankful to reviewer No.3 (#R3).

The reviewer’s’ comments have definitely helped us to improve the quality of the manuscript.

  1. English should be checked, and besides grammar, terminology should be carefully revised. In figure 2 spelling of anatase, rutile and tenorite is not English

We appreciate your comment! Fig.2 was replaced with a new one, where the correct English names of the compound were used. See section 3.1, please.

2. When describing FTIR results, instead of deformation and valence vibrations one should write bending and stretching, and maybe not only.

We corrected the terminology according to your suggestion. Thank you so much for your comments!

3. Presenting the data of measurements, one should take into account the accuracy of methods. The specific surface area can hardly be measured with the precision of 1%, and if the instrument shows the value of, say, 2,7061 m2/g, as in Table 1 (5 meaningful figures!), one has to round it up to at least 2.70, or better 2,7. Even the mass of the probe cannot be measured better than +/- 0.1 mg! The same question could be put about lattice parameters in the table or the data on the composition of samples shown in Table 2. Maybe, some words should be said about the accuracy in the text, if the authors find that the last figures in these data are necessary.

We appreciate your comments about both BET and XRD expression of results! We corrected BET values. Check Table 1, please. We keep the expression of lattice parameters according to our experiences to publish refinement XRD data to structurally related journals (see Ref.27 and 28, please). The idea to keep such accuracy is coming from the requirement to check how the original TiO2 lattice is distorted after the introduction of any dopant. Sometimes, the distortions are negligible, but even 0.1-0.2% are enough to shake up the electronic properties of the whole doped system. We corrected the % of the composition by EDS (see Table 2, please).

4. The authors used samples doped by 2% of Cu, however, it was shown that maximum photocatalytic activity for titania doped with Al, Sc or V is reached at much lower dopant concentrations and then seriously decreases (see, e.g., Murashkina et al, J. Phys. Chem. C, 119(2015), no.44, p.24695 or Murzin et al, Top. Catal., 2020, p.1, and refs therein). Why not at least discuss the possible dependence of activity on the amount of dopant?

We thank the reviewer for such a great idea to provide a more relevant discussion focused on the activity of the Cu-TiO2 system considering the concentration of the Cu dopant. We appreciate the significance of this suggestion and updated the discussion by comparing our results and yet published outcomes from other authors. New references in this context were used and added to the list of references.

Yoong et al Development of copper-doped TiO2 photocatalyst for hydrogen production under visible light, Energy, 34, 2009, 1652-1661

Bandara et al. Highly stable CuO incorporated TiO2catalyst for photocatalytic hydrogen production from H2O, J Photochem. Photobiol. Sci., 2005, 4, 857K.Y. Song et al. Photocatalytic activity of Cu/TiO2 with the oxidation state of surface- loaded copper, Bull. Korean Chem. Soc., 20 (1999), 957G. Colon, Towards the hydrogen production by photocatalysis, Appl. Catal. A: General, 518, 2016, 48.G. Colon et al. Cu-doped TiO2 system with improved photocatalytic activity, Appl Catal Environmental, 67, 2006, 41

Some misprints, certain drawbacks, or typical mistakes:

1) The abbreviation MRs first appeared in the abstract, line 31, before it is explained in the text (line 78). Why not do it at first appearance in line 20? – Yes. You are right. We corrected this issue according to your suggestion,

2) line 110:  he lyophilized precursor – we corrected the spelling.

3) line 153: This program permits… better  This program allows or enables -Yes. We replaced permits. 

4) lines 229-230: the lattice parameter a decrease and the parameter c increase with increasing of temperature. better: the lattice parameter a decreases and the parameter c increases with the increasing temperature. Thank you for your correction. We improve English grammar.

5) line 267:  mesoporous annealing should be   mesoporous, annealing  without comma no sense; - Yes. We corrected this line due to give better sense.

6) line 270: pore radius of … samples increase  should be  pore radius of … samples increases... Thank you! We improve the English grammar.

7) lines 554, 556 and not only:   instead typical for… better  typical of Thank you! We improve English grammar.